# Experimental Study Comparing the Effectiveness of Physical Isolation and ANN Digital Compensation Methodologies at Eliminating the Stress Wave Effect Error on Piezoelectric Pressure Sensor

**DOI:** 10.3390/s20082397

**Published:** 2020-04-23

**Authors:** Lei Feng, Tiehua Ma

**Affiliations:** 1School of Electrical and Control Engineering, North University of China, Taiyuan 030051, China; 2Key Laboratory of Instrumentation Science & Dynamic Measurement, Ministry of Education, North University of China, Taiyuan 030051, China

**Keywords:** piezoelectric pressure sensor (PPS), stress wave effect (SWE), split Hopkinson pressure bar (SHPB), physical isolation, digital compensation, artificial neural network (ANN)

## Abstract

Stress wave, accompanied by explosion shock wave overpressure measurement and dynamic pressure calibration on shock tube, could cause error signals in the piezoelectric pressure sensor (PPS) used for measuring and calibrating. We may call this error the stress wave effect (SWE). In this paper, the SWE and its isolation from PPS were studied by using a split Hopkinson pressure bar (SHPB). In the experimental study of SWE, when increasing the input stress, the corresponding output signal of the PPS was analyzed, and the existence of SWE was verified using the result of the spectrum analysis of the output signal. The stress wave isolation pedestal used in the stress wave isolation experiment was made of nylon and plexiglass polymer materials. The effects of the isolation pedestal’s materials and length on the stress wave isolation were analyzed using the study results. Finally, an artificial neural network (ANN) was trained with the data of the SWE study and was further applied to compensate the SWE error of the PPS output signal. The compensating results were compared with the isolating results, and the advantages and disadvantages of the digital compensation and physical isolation methods were analyzed.

## 1. Introduction

In the military industry, the explosion shock wave overpressure measurement has often been used to evaluate the power of ammunition. However, to increase the damage, there are fragments, steel balls, tungsten beads, and other objects in the actual warhead, and these objects usually arrive at the measurement device faster than the shock wave. Therefore, shock wave and fragments often hit the measurement device, sensor, or mounting structure, resulting in abnormal measurement data [1,2,3]. As the material of the fragments is usually made from metals with great hardness, when the fragments hit the metal structure of the measurement device, they cause the device to vibrate or produce the stress wave propagating in the device. The two interference sources, vibration and stress wave, will cause the abnormal overpressure measurement signal.

In particular, piezoelectric pressure sensor (PPS) is intended to measure the overpressure of shock wave in the air, so signals caused by other reasons can be regarded as interference to the overpressure signals. As shock waves and fragments have limited impact energy on the PPS, the stress wave generated by the impact can be considered as a sound wave in the solid rather than a shock wave. This is because generating a shock wave requires disturbance in the medium behind to keep up with the disturbance ahead. Evidently the impact will not produce a shock wave in a solid.

In addition, when the shock tube is used to dynamically calibrate the pressure sensor, it will produce vibration or stress wave propagating in the tube after the diaphragm bursts, which will also cause interference to the pressure sensor and produce an abnormal signal [4,5]. The reason why shock tubes can produce vibration or stress waves may be that the broken part of the diaphragm or shock wave hits the tube’s wall or ends after the diaphragm bursts [6]. The exact reason is still unclear. Considering that most PPSs have an acceleration-compensated function [1], this study only focused on the negative affect of stress waves on their measurement signal and a way to exclude them from the result.

The current studies in this field have mainly focused on the errors caused by high-g impact and vibration to the PPS output signal. Fujing Xu studied the acceleration-induced effects of PPS experimentally and used system identification theory to model the acceleration effect [7]. C. Gradolph studied the piezoresistive pressure sensor’s performance when used in high-g and high vibration environments [8].

In this paper, the authors first analyzed the mechanism of stress wave on the pressure sensor, and then carried out experimental research based on this theoretical basis. The experiment was completed on the split Hopkinson pressure bar (SHPB) device, and the existence of stress wave effect (SWE) was verified by analyzing the data collected from the experiment. Furthermore, the authors studied the effect of the isolation pedestal’s material and size on the SWE elimination. Finally, the SWE error compensation model was built by an artificial neural network (ANN), and the SWE experimental data were used to train and obtain the error compensation model. The results after compensation were compared with the isolating results, and the advantages and disadvantages of the two methods were analyzed.

## 2. Mechanism of Stress Wave Acting on Piezoelectric Pressure Sensor (PPS)

### 2.1. Explosion Shock Wave Overpressure Measurement

The overpressure measurement of an explosion shock wave is usually performed at a distance from the detonation center using a piezoelectric or piezoresistive pressure sensor as a pressure sensing unit. According to the theoretical model, the pressure sensor can be equivalent to a second-order mechanical system composed of mass block, damping, and spring. When the sensor is exposed to pressure, it can produce a charge or voltage value corresponding to the pressure [9,10,11]. Therefore, it can be understood that the blast pressure is transferred from the detonation center to the sensor using air as a medium, namely the transfer of gas pressure to the solid. The schematic diagram of a typical overpressure measurement of explosion shock wave is shown in Figure 1a. There are two types of incident ways of the shock wave. One is the ground incident mode, where the sensitive surface of the pressure sensor is flush with the ground level, and the overpressure value of the Mach wave formed by the interaction between the incident and the reflected shock wave is measured. The other is the air incident mode, where a pen-shaped pressure sensor (such as ICP^®^137B2XB produced by PCB^®^) is used to measure the incident shock wave overpressure in the air. Regardless of which incident mode is used, there is a high probability that shock wave and explosive fragments will hit the mounting plate or the pen-shaped sensor’s housing and cause a stress wave propagating in the sensor, and the stress wave greatly disturbs the overpressure signal. As shown in Figure 1b, the left curve is the theoretical overpressure curve of the shock wave and the right one is the overpressure curve obtained in an actual explosion. It is clear that the curve in the actual explosion is consistent with the theoretical one, but there is also a great difference between them. In particular, before the edge of the curve rises, the actual one had a large oscillation near the baseline that is most likely to have been caused by the stress wave produced by the explosive fragments hitting the metal structure before the shock wave arrives at the measurement device.

Figure 1c shows the internal structure diagram of the PCB^®^ PPS products [1]. The ‘Quartz Plates’ in Figure 1c refer to the three piezoelectric quartz plates between the electrodes. The unmarked part in Figure 1c between the diaphragm and the first quartz plate is a mechanism with a rigid material to transfer pressure from the diaphragm to the quartz plates. As shown in the figure, the sensor has an acceleration-compensated function. The basic mechanism is that an acceleration-sensing crystal is added to the sensor to measure the acceleration, and the charge generated by the crystal will offset the negative effect of the acceleration to pressure signal. The piezoresistive pressure sensor has no acceleration-compensated function. However, due to the small volume, light mass, and large stiffness of the piezoresistive silicon diaphragm, the piezoresistive pressure sensor has a low response to acceleration. The acceleration sensitivity of the Kulite^®^ piezoresistive pressure sensor is generally below the order of 3×10^−5^% FS/g. In contrast, the acceleration sensitivity of the PPS after compensation is generally around 10^−3^% FS/g due to its large volume and mass [12,13].

### 2.2. Mechanism of Stress Wave

#### 2.2.1. One-Dimensional Stress Wave Propagating Mechanism

First, this section discusses the wave equation of stress wave in a solid hit by a solid particle and made two basic hypotheses: (1) the cross section of the stress wave remains flat when it propagates, and there is only uniformly distributed axial stress along the cross section; and (2) stress is a unary function of strain, that is, the constitutive relation of material is $\sigma =\sigma \left(\varepsilon \right)$. The basic equations include the continuity equation (mass conservation equation), motion equation (momentum conservation equation), and constitutive relation of material (physical property equation) [14].

Mass conservation equation:(1)$$\frac{\partial v}{\partial X}=\frac{\partial \varepsilon }{\partial t}$$

Momentum conservation equation:(2)$$\rho_{0}\frac{\partial v}{\partial t}=\frac{\partial \sigma }{\partial X}$$

Constitutive relation equation:(3)$$\sigma =\sigma \left(\varepsilon \right)$$

In general, the constitutive relation equation is continuous and differentiable. Assuming that the first derivative is a non-zero positive number, introducing *C_0_* satisfies:(4)$$C_{0}^{2}=\frac{1}{\rho_{0}}\frac{d\sigma }{d\varepsilon }\;\frac{d\sigma }{d\varepsilon }>0$$

Eliminate σ or ε, and get:(5)$$\frac{\partial^{2}u}{\partial t^{2}}-C_{0}^{2}\frac{\partial^{2}u}{\partial X^{2}}=0$$

In Equations (1)–(5), *v* is the velocity of the particle, *X* is the coordinates of the particle, ε is the strain, *t* is the time, $\rho_{0}$ is the density, σ is the stress, *C_0_* is the wave velocity, and *u* is the particle displacement. Equation (5) is the wave equation based on the above two hypotheses. If hypothesis 1 is not true, then there is not only axial stress in the cross section, but also transverse stress, which is essentially caused by the transverse motion of the particle. Due to the Poisson effect, the wave equation becomes:(6)$$\frac{\partial^{2}u}{\partial t^{2}}-\mu^{2}r_{g}^{2}\frac{\partial^{4}u}{\partial X^{2}\partial t^{2}}=C_{0}^{2}\frac{\partial^{2}u}{\partial X^{2}}$$

In Equation (6), μ is Poisson’s ratio, and *r_g_* is the rotating radius of the section against the axis. The second term in Equation (6) represents the transverse effect. It can be seen that the stress wave velocity is no longer constant *C_0_*, and the harmonics of different frequencies *f* or wavelength λ will propagate at different velocities *C*. Assume a cylinder with radius *a* to study, so rg=a2 and the relation between *C* and *C_0_* is as follows:(7)$$\frac{C}{C_{0}}\approx 1-\mu^{2}\pi^{2}{\left(\frac{a}{\lambda }\right)}^{2}$$

It can be concluded from Equation (7) that the higher the frequency, the shorter the wavelength, so the value of Equation (7) and the wave velocity will become smaller. The lower the frequency, the longer the wavelength, so the value of Equation (7) and wave velocity will become larger. For the linear elastic wave, it can be superimposed by harmonic components of several different frequencies, so in the stress wave’s propagating process, the waveform will spread out, which is called geometric dispersion. However, the transverse stress can be ignored when the wavelength λ is 6~10 times larger than the diameter *a* of the cylinder, so the wave velocity is constant in the same medium.

#### 2.2.2. Reflection and Transmission Mechanism of Stress Wave

In the actual propagation process, stress waves often encounter many kinds of medium, whose material and geometric dimensions are different. Therefore, it is necessary to discuss the reflection and transmission mechanism of the stress wave at the interfaces of the different mediums. For two different mediums, the product of the density and wave velocity is defined as wave impedance ρC, while the product of the wave impedance and their interface’s area is defined as generalized wave impedance ρCA. When incident stress wave $\sigma_{I}$ passes through section $A_{1}$ of medium 1 and enters section $A_{2}$ of medium 2, reflection and transmission will occur simultaneously. The reflected stress $\sigma_{R}$ and transmitted stress $\sigma_{T}$ are respectively:(8)σR=FσI       σT=TσIA1A2

The reflection coefficient *F* and transmission coefficient *T* are as follows:(9)$$F=\frac{1-n}{1+n}\;T=\frac{2}{1+n}$$

In Equation (9), *n* is the ratio of the two kinds of medium’s generalized wave impedance, and $n=\frac{{\left(\rho CA\right)}_{1}}{{\left(\rho CA\right)}_{2}}$. As shown in Figure 1c, the PPS contained multiple components and each of the components was made of a different material. Therefore, when the stress wave acts on the sensor, there must be reflection and transmission.

Piezoelectric or piezoresistive pressure sensors have an internal structure that is very complex and includes many functional components and varying sizes. Therefore, the stress wave and its acting mechanism are not as simple. It is necessary to consider the influence of the different amplitudes and changing rates of the stress wave, and use stress waves with different amplitudes and changing rates in the specific tests. Therefore, the experimental research scheme of the interaction between stress wave and PPS can be preliminarily determined, mainly based on the following aspects:In actual test applications, stress waves enter the sensor from its side. That is, in the overpressure measurement of explosive shock waves, the stress wave is laterally introduced when the sensor is installed on the metal disk [2]. When the pressure sensor is dynamically calibrated on the side wall or at the end of the shock tube, the stress wave is also laterally introduced.Under different incident stress wave conditions (amplitude and changing rate), the output responses of the PPS can be studied.

## 3. Experimental Study on Stress Wave Effect (SWE)

### 3.1. Split Hopkinson Pressure Bar (SHPB) Equipment and Experimental Scheme

SHPB equipment is widely used to study the dynamic mechanical properties of materials under impact load [15,16,17], which is characterized by the ability to obtain rapidly changing stress waves. In many cases, SHPB equipment is particularly used in the study of building materials such as concrete, asphalt, foam buffer materials, etc. Therefore, the cross-section area of the bar is very large. There have also been reports where SHPB has been used to study the stability of pyrotechnics [18]. To summarize, SHPB is a kind of widely used high impact test equipment.

Figure 2a shows the schematic diagram of the SHPB. Its basic working principle is to make the projectile obtain a certain speed using the air gun to then hit the input bar. The stress wave generated by the impact propagates along the input bar to the output bar. The input bar is directly in contact with the output bar, and lubricating grease is smeared on the contact surface of the two bars to fill the tiny gap. Since the two bars are made of steel and have the same cross-section, the stress waves do not reflect between their contact surface (Equations (8) and (9)) and the lubricating grease will not weaken the stress waves. The values of the stress waves in the two bars are approximately equal and the range of the strain rate is 10^2^–10^4^ s^−1^ [19]. The PPS is mounted laterally on the output bar. The shape of the pressure bar is generally slender to suppress the dispersion of stress wave, that is, the ratio of length l to diameter *d* of the bar is large: l≫d. In general, the way SHPB works to test the dynamic mechanical properties of the materials is to clamp the material sample between the input and output bars.

Strain signals in the two bars are measured by strain gauges on the bar, respectively. Strain gauges are glued to the bar and used to record the incident, reflected, and transmitted strain signals. In conventional dynamic mechanical properties testing of the materials, a strain gauge on the input bar is used to record the incident and reflected strain signals and the strain gauge on the output bar is to record the transmitted strain signals through the specimen [20]. In this paper, the strain gauge on the output bar (strain gauge 2) was used to record signals in the SWE experiment. The reason why two strain gauges are not used to record simultaneously is that the PPS itself can produce output signals induced by the stress. This is not the same as in the conventional material mechanics test.

Then, the stress wave in the bar is calculated through strain signals based on Hook’s law and used to study the response of the material sample. Strain gauges and hyper dynamic strain instruments are used to measure the stress wave signals in bars. The output charge signal of the PPS is converted into a voltage signal through the charge amplifier. These two kinds of signals are collected by the data acquisition card (DAQ card) and then stored and analyzed by the computer. It is important to note that the PPS is mounted on the output bar at 200 mm from the right end through a blind threaded hole. As shown in Figure 2a, the sensitive surface of the PPS is exposed to the space with atmosphere, but not in contact with the output bar. This ensures that only the stress wave from the PPS wall, but no gas pressure change, is put into the PPS.

Generally, the hit bar of the SHPB test equipment is the projectile fired by the air gun, so the impact velocity is usually high and the stress wave amplitude generated is also large. However, the purpose of this experiment was to study the response of the PPS to the stress wave. The output signal must be obtained on the premise that the PPS is not damaged by the stress wave. Therefore, the velocity of the hit bar should not be too high.

According to one-dimensional stress wave propagation theory, the stress amplitude of the elastic wave in the bar increases directly with the increase of impact velocity. If the dynamic yield limit of the material under one-dimensional stress is *Y*, plastic deformation will occur in materials when the impact velocity v is greater than the yield velocity $v_{Y}$, namely [14]:(10)$$\left|v\right|>v_{Y}=\frac{Y}{\rho_{0}C_{0}}$$

The piezoelectric element of the PPS used in this experiment was quartz crystal. According to the relevant study, the related mechanical parameters of the quartz crystal are as follows: dynamic yield limit *Y =* 300 MPa, wave impedance $\rho_{0}C_{0}$
*=* 1.3 × 10^7^ kg/m^2^s–1.9 × 10^7^ kg/m^2^s, the yield limiting velocity calculated is $v_{Y}$
*≈* 15.9 m/s–22.6 m/s. Therefore, in order to protect the quartz crystal of the PPS from plastic deformation, the impact velocity must be controlled below 15.9 m/s. Due to the limitation of the speed and caliber of the air gun, it is not suitable to use the air gun to fire the hit bar. The hit bar is loaded by releasing from a certain height. The support rod is released at a certain angle to the vertical direction. The greater the angle, the greater the speed of the hit bar, as shown in Figure 2b. The actual photograph of the SHPB equipment is shown in Figure 2c.

### 3.2. Experimental Data Analysis

The hit bar was released from different angles: 15, 30, 45, 60, 75, 90, 105, 120, 135, 150, 165, and 180 degrees, successively, therefore, there were 12 releasing angles, that is, 12 kinds of stress wave input to the PPS. As the releasing angle increased, the amplitude of the input stress wave also gradually increased. The single experimental data at 30, 60, 90, 120, 150, and 180 degrees were selected and drawn in Figure 3. In Figure 3, the stress signals were collected by the strain gauge on the output bar (gauge 2). The unit of the stress wave signal is in MPa and the output signal of PPS can be considered as equivalent pressure (EP), whose unit is also in MPa.

With the increase in the releasing angle of the hit bar, the maximum peak value of the stress wave signal increased and the peak value of the PPS output signal also increased correspondingly. The stress wave signal and PPS output signal were both positive and negative. The curve presented damped oscillations related to the reflection to the bars’ end. In addition, the output signal of the PPS had some degree of drift, and its baseline moved down significantly at 120 degrees. The data from one SWE experiment are listed in Table 1. In Table 1, the highest positive peaks and highest negative peaks represent the maximum positive and negative absolute values on the stress wave and PPS output curves; the spectrum points represent the frequency points corresponding to the first two maximum values on the spectrum curves of the two type signals after fast Fourier transform (FFT); and the positive EP–stress ratio represents the ratio of the positive EP value to the positive stress wave value in the same column in the table and the same calculating method for the negative EP–stress ratio.

According to the position of the strain gauges on the bars and the velocity of the stress wave in steel (5800 m/s), the frequency of stress wave signal can be calculated theoretically. The geometric position relationship between the strain gauges and the input and output bars is shown in Figure 4. When the stress wave passes through strain gauge 2 for the first time, it will propagate to the right end of the bar. The stress wave then passes through strain gauge 2 again after reflection and then propagates to the left. The stress wave continues to propagate back and forth in the bars, and its amplitude gradually decreases.

The distances between strain gauge 2 and the two ends (right and left) of the bar were 0.27 m and 0.53 m, respectively. The distance the stress wave passed through the strain gauge twice before and after was twice as much as that between the strain gauge and the bar’s end, namely, the distances were 0.54 m and 1.06 m, respectively. The frequencies calculated from these distances and the stress wave velocity were 10.74 kHz and 5.47 kHz. Taking the 120 degrees releasing angle as an example, the spectrum analysis of the stress wave signal and PPS output signal was completed, and the results are shown in Figure 5. It can be seen from the figure that the energy of the stress wave signal was concentrated at two frequencies of 5.455 kHz and 10.91 kHz, and the PPS output signal was concentrated at 5.091 kHz and 11.48 kHz. The concentrating frequency point of the output signal was very close to the calculated frequency point, which can prove that the stress wave is the cause of the output signal of the PPS.

We repeated the test five times at each releasing angle and drew the trend of the average values calculated from the five sets of data in Figure 6. The error bar on the curve represents the confidence interval when the statistical sample size was *n =* 5 and confidence level was *P =* 0.95. Figure 6a is the highest positive and negative peak curves of stress wave and (b) is the peak curves of the PPS output. Figure 6c is the EP–stress ratio curve, and the EP–stress ratio refers to the ratio of positive and negative peaks corresponding to the PPS output signal and stress wave signal. Since the two kinds of signals have the same dimension MPa, the dimension of the EP–stress ratio is one. In Figure 6, S+ and P+ respectively represent the positive peak curve of the stress and EP, and T+ is the positive peak curve of the EP–stress ratio. S-, P-, and T- represent the corresponding negative values of the three variables. We can see that the positive and negative peak values of the stress, EP, and EP–stress ratio increased with the increase in releasing angle. When the releasing angle reached a maximum of 180 degrees, the positive and negative peaks of the PPS output reached a maximum of 1.2 MPa, while the range of the PPS used in this test was only 10 MPa.

The following conclusions can be drawn from the analysis of the SWE experimental data. (1) In the absence of external pressure acting on the sensitive surface of the PPS, the stress wave causes the PPS to output a signal. This signal is similar to the stress wave signal in the frequency domain, both have positive and negative, and show a tendency of oscillating attenuation. (2) As the peak value of the stress wave increases, the peak value of the PPS output also increases correspondingly. The maximum PPS output value in the experiment exceeded 10% of its full range and may continue to increase as the stress increases until the sensor is damaged. (3) The EP–stress ratio increases with the increase in stress value, which proves that the efficiency of the stress wave affecting sensor is enhanced.

## 4. Experimental Study on Stress Wave Isolation

From the SWE experimental results, we can see that the stress wave that laterally entered the PPS could make it produce a certain output signal, which is a kind of interference compared with the normal pressure signal. In addition, when the amplitude of the stress wave reaches a certain degree, the sensor will be damaged. Taking the above two factors into consideration, the negative effect of the stress wave on the PPS should be minimized.

### 4.1. Experiment Scheme of Stress Wave Isolation

The basic principle of stress wave isolation is to increase its reflection on the interface of different media, thus reducing the stress wave entering into the PPS. According to the stress wave propagation theory above-mentioned, the factors affecting transmissivity and reflectivity of the stress wave are the wave impedance and contact area of the medium. An effective stress wave elimination method is to add a material with a small wave impedance outside the thread of the PPS as the isolation pedestal. The PPS is threaded to the isolation pedestal and the isolation pedestal must also be easy to install on the external structures such as a metal plate and a shock tube wall. In addition, special attention should be paid to the fact that the material of the isolation pedestal must have a large stiffness, so that the gas flow field remains stable without distortion as the shock wave flows by.

Taking all this into consideration, the isolation pedestal is made of nylon and plexiglass polymer materials. We also used the SHPB equipment in Section 3.1 as the experimental device, but the difference in this isolation experiment was that the PPS was installed on the isolation pedestal that was clamped between the input bar and the output bar. The strain gauge on the input bar (strain gauge 1) was used to record signals in the isolation experiment. The local structure diagram of the experiment device is shown in Figure 7a. Two kinds of isolation pedestals with different lengths (16 mm and 30 mm) were used for a comparative analysis. The isolation pedestal was made into a rectangular shape and the threaded hole was drilled in the middle to install the PPS. The specific dimensions and photographs of the isolation pedestals are shown in Figure 7b,c.

As shown in Figure 7a, the input bar and the output bar held the isolation pedestal on the two 16 mm × 12 mm surfaces, that is, the stress wave entered into the isolation pedestal from this surface. According to the above-mentioned Equations (8) and (9) for the reflection and transmission of stress waves at different media interfaces, it can be seen that when the stress waves were transferred from a medium with a large cross-section area to a medium with small one, the amplitude of transmitting stress increased when compared to that with the equal cross-section area. Based on this conclusion, the input and output bars were a round bar with a diameter of *Φ* = 20 mm, a cross-section area of 314 mm^2^, and the isolation pedestal’s lateral area was 192 mm^2^. It is clear that the strength of the transmitting stress wave increased after passing through this contact surface. When setting up the experimental scheme, we took factors such as contact area, material, difficulty in manufacturing, and convenience in installing into consideration. The contact area of the two isolation pedestals with different materials was set equal. Therefore, only the wave impedance and length may affect the experimental results when comparing the nylon and plexiglass materials. The wave impedance parameters of each material in the experiment are shown in Table 2.

### 4.2. Experimental Data Analysis

As in the SWE experiment, releasing the hit bar from different angles with the minimum of 15 degrees and the maximum of 180 degrees, the angle increased by 15 degrees successively, so there were 12 releasing angles. We repeated the experiment five times at each releasing angle and obtained five sets of experimental data. The single experimental data of four types of isolation pedestals at 120 degrees were selected and are drawn in Figure 8.

Compared with the un-isolating data in Figure 3, it can be preliminarily seen that the frequency of the oscillation decreased, especially the nylon isolation pedestal. In addition, the positive and negative peaks of the curves also decreased significantly. The output of the PPS of the nylon pedestal isolation experiment decreased more significantly than the Plexiglass pedestal when the length of the two types of pedestal were equal. The partial positive peak of the 30 mm Plexiglass pedestal isolation experimental results was larger than the positive peak of the SWE experimental results at 120 degrees. The experimental data of the four isolation pedestal are analyzed below.

As in the SWE experiment, the trend of the average values calculated from the five sets of data are plotted in Figure 9 and Figure 10, respectively, for the nylon and Plexiglass isolation pedestals. The error bar on the curve represents the confidence interval when the statistical sample size was *n =* 5 and the confidence level was *P =* 0.950.

Figure 9a and Figure 10a are the positive and negative maximum peak curves of the stress wave, respectively. Figure 9b and Figure 10b are the maximum peak curves of the PPS output after isolation. Figure 9c is a larger view of Figure 9b. Figure 9d and Figure 10c are the EP–stress ratio curve after isolation.

In Figure 9 and Figure 10, SWE+ and SWE− represent the positive and negative peak values (stress, EP, and EP–stress ratio, respectively) of the SWE experiment for comparison. 

In Figure 9, Nylon16+ and Nylon16− represent the positive and negative peak values (stress, EP, and EP–stress ratio, respectively) of the 16 mm nylon isolation pedestal experiment; Nylon30+ and Nylon30− represent the positive and negative peak values (stress, EP, and EP–stress ratio, respectively) of 30 mm nylon isolation pedestal experiment. The results showed that the stress, EP, and EP–stress ratio peak values of the nylon isolation pedestal experiment decreased sharply.

In Figure 10, Plexi16+ and Plexi16− represent the positive and negative peaks value (stress, EP, and EP–stress ratio, respectively) of 16 mm Plexiglass isolation pedestal experiment; Plexi30+ and Plexi30− represent the positive and negative peak values (stress, EP, and EP–stress ratio, respectively) of the 30 mm Plexiglass isolation pedestal experiment. The results showed that the stress peak values of the Plexiglass isolation pedestal experiment decreased sharply and the EP peak values of the 16 mm Plexiglass isolation pedestal experiment decreased sharply. However, the peak value of 30 mm Plexiglass isolation pedestal did not show the same trend and was even greater than the peak value of the SWE experiment. The EP–stress ratio value of the Plexiglass isolation pedestal experiment was bigger than that of the SWE experiment. This indicates that the Plexiglass isolation pedestal does not have an isolating function.

In addition, the experimental data showed no obvious changing trend with the length of the nylon and Plexiglass isolation pedestal.

The following conclusions can be drawn from the analysis of the stress wave isolation experimental data: (1) The output signal amplitude of the PPS after isolation by the nylon and Plexiglass isolation pedestal was significantly reduced. (2) The EP–stress ratio of nylon isolation pedestal *T_N_* was small and the EP–stress ratio of the Plexiglass isolation pedestal *T_P_* was big, and the relation between the two isolation EP–stress ratios and the EP–stress ratio of SWE *T* was: *T_N_**<**T**<**T_P_*, so Plexiglass is not suitable to make the isolation pedestal. (3) The EP–stress ratio of the nylon isolation pedestal showed no obvious change trend with its length, but the EP–stress ratio of the longer Plexiglass isolation pedestal was bigger than the short one. (4) The nylon isolation pedestal can effectively filter out the high frequency components of the PPS output signal.

## 5. SWE Error Compensation Based on an Artificial Neural Network (ANN)

As a recent research hotspot, ANNs have the characteristics of high parallelism, strong nonlinear approximation, adaptive and self-learning, etc. As such ANNs have been widely applied in many fields [21,22,23]. This study made full use of the nonlinear mapping function of the ANN, and designed an error compensation model of SWE without knowing the system model and parameters of the PPS. The data verification results showed that the error compensation method based on an ANN can effectively reduce the SWE error and provide a feasible digital signal processing method to eliminate the effect of the SWE.

### 5.1. Artificial Neural Network (ANN) Compensation Model

In general, PPS can be considered as a linear time-invariant system, whose dynamic characteristics can be expressed by linear n-order ordinary differential equations with constant coefficients:(11)$$a_{n}\frac{d^{n}y}{dt^{n}}+a_{n-1}\frac{d^{n-1}y}{dt^{n-1}}+\cdots +a_{1}\frac{dy}{dt}+a_{0}y=b_{m}\frac{d^{m}x}{dt^{m}}+b_{m-1}\frac{d^{m-1}x}{dt^{m-1}}+\cdots +b_{1}\frac{dx}{dt}+b_{0}x$$

In Equation (11), *x = x(t)* is the input signal and *y = y(t)* is the output signal, *a_i_ (i =* 0,1,2...n), and *b_j_ (j =* 0,1,2...m) is composed of various physical parameters related to the internal structure and materials of the PPS. The order of the equations is determined by the structure and working principle of the PPS. In general, the PPS can be approximated as a second-order system, so when *n =* 2, *m =* 0, Equation (11) can be converted into:(12)$$a_{2}\frac{d^{2}y}{dt^{2}}+a_{1}\frac{dy}{dt}+a_{0}y=b_{0}x$$

As the ANN model is to compensate the stress wave response signal of the PPS, the error caused by this SWE needs to be eliminated. Then, the input of the PPS system is the stress wave signal *x(t)*, and the response of the PPS to stress wave is *y(t)*. After ANN compensation, the system output should be *0(t)*. The basic compensation principle is shown in Figure 11a.

The algorithm was designed based on the basic compensation principle and its structure is shown in Figure 11b. In the figure, *S* represents the differential operator. The output signal *y* is the stress wave response signal of the PPS. An *S* operation on the *y* signal corresponds to one derivative operation, and the derivative result of each order of *y* are successively put into the ANN. Let the output of ANN be *0(t)*, that is, the error signal caused by stress wave is reduced to zero, and the parameters of the ANN are trained. Using a back propagation (BP) neural network with a simple structure, but given its disadvantages such as easy to fall into the local minimum and slow training convergence speed, genetic algorithm (GA) is used to improve the BP neural network (BPNN). The specific methods are to optimize the initial weights and threshold coefficients. The improved BPNN can rapidly converge to the global optimal solution [24].

### 5.2. Model Training and Result Analysis

BPNN adopts a three-layer network structure, in which the ratio of the number of neuron nodes in the input layer, the middle layer, and the output layer is 3:5:1. As above-mentioned, PPS can generally be considered as a second-order system, so the number of neuron nodes in the input layer was set as three. An appropriate number of nodes in the middle layer can not only improve the calculation accuracy, but also reduce the training cost. Therefore, it is appropriate to set the number of nodes in the middle layer as five. Training datasets were obtained by time discretization and normalization of the two derivatives of the SWE experimental data. That is, $\left\{y,\dot{y},\ddot{y}\right\}$ is the input training dataset of the network and $\left\{0\right\}$ is the output training dataset of the network. After 300 iterations, the results converged, and the network residual error reached the minimum.

Partial SWE experimental data were used as the verification dataset of the network and the verification results are drawn in Figure 12. Figure 12a shows the comparison of the PPS response data to the stress wave (EP), the isolation experimental data (EPi), and the BPNN error compensation data (EPc) when the releasing angle was 120 degrees. Figure 12b shows the comparison of the isolation experimental data and the BPNN error compensation data when the releasing angle was 120 degrees. Figure 12c,d show the comparison of the maximum peak lines of the above EP, EPi, and EPc at different releasing angles.

As shown from Figure 12b,d, the BPNN error compensation result was almost zero at each releasing angle and the amplitude was significantly smaller than that of the isolation experimental data, which is an ideal off-line SWE error compensation algorithm. By comparing the advantages and disadvantages of the isolation pedestal and the BPNN error compensation, we can conclude that the isolation pedestal, as a physical method to isolate the SWE, was used in the experiments conveniently and could protect the PSS from damage. However, the amplitude of the SWE isolation result was larger than the BPNN error compensation. The error compensation method based on BPNN is an off-line digital signal processing method. Although the error amplitude of the compensation method is better than that of the isolation method, its outstanding disadvantage is that it cannot protect the PPS from the damage caused by the stress waves and it is difficult to use online in embedded electronic instruments with limited hardware configurations. Generally speaking, the acceptable signal error range in engineering is less than 5%, so the relationship between the error range and engineering practicality should be weighed to choose one of the above two SWE error elimination methods.

## 6. Conclusions

In this paper, the SWE of PPS was studied by experimental methods. First, the stress wave was directly put into the PPS, and the maximum output value of the PPS could reach up to 12% of the full range. Furthermore, with the increase in the stress amplitude, the EP–stress ratio also increased correspondingly. Second, the isolation effect of the polymer material isolation pedestal on the stress wave was studied. The experimental data showed that the isolation effect of the nylon isolation pedestal was good. Finally, the digital error compensation method based on an ANN was adopted to study the error compensation of the SWE. The compensation effect by this method was better than the isolation method in error elimination, but its deficiency was also obvious. The advantage of physical isolation is that it can protect the PPS from irreversible damage and improve the survivability of the sensor in stress shock. In engineering, a SWE error elimination method can be selected by considering the factors of acceptable error and practicality.

## Figures and Tables

**Figure 1 sensors-20-02397-f001:**
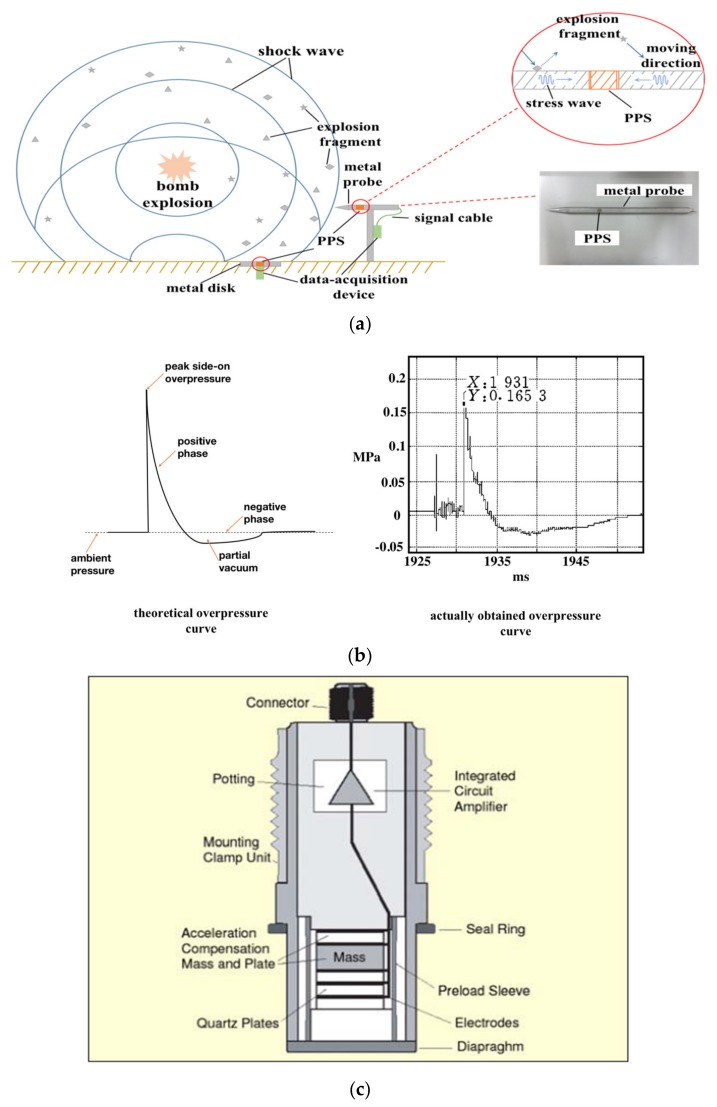
Schematic diagram of a typical explosion shock wave overpressure measurement: (**a**) side on overpressure measurement schematic diagram; (**b**) theoretical and actual side on overpressure curves; (**c**) internal structure of the piezoelectric pressure sensor (PPS).

**Figure 2 sensors-20-02397-f002:**
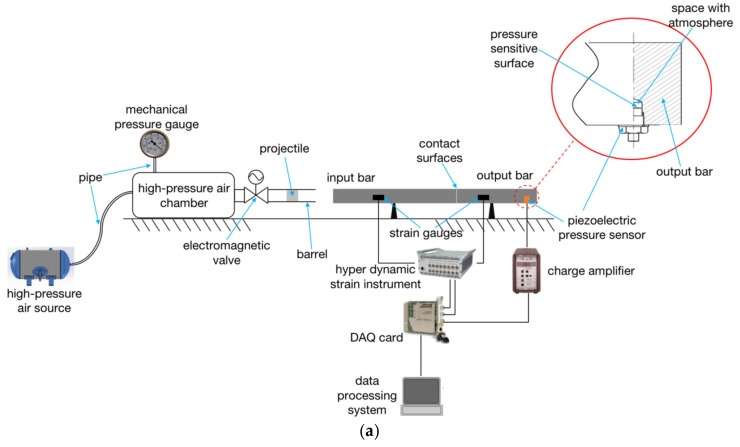
The split Hopkinson pressure bar (SHPB) test equipment. (**a**) Structural schematic diagram. (**b**) Loading way of stress wave. (**c**) Photograph of the SHPB.

**Figure 3 sensors-20-02397-f003:**
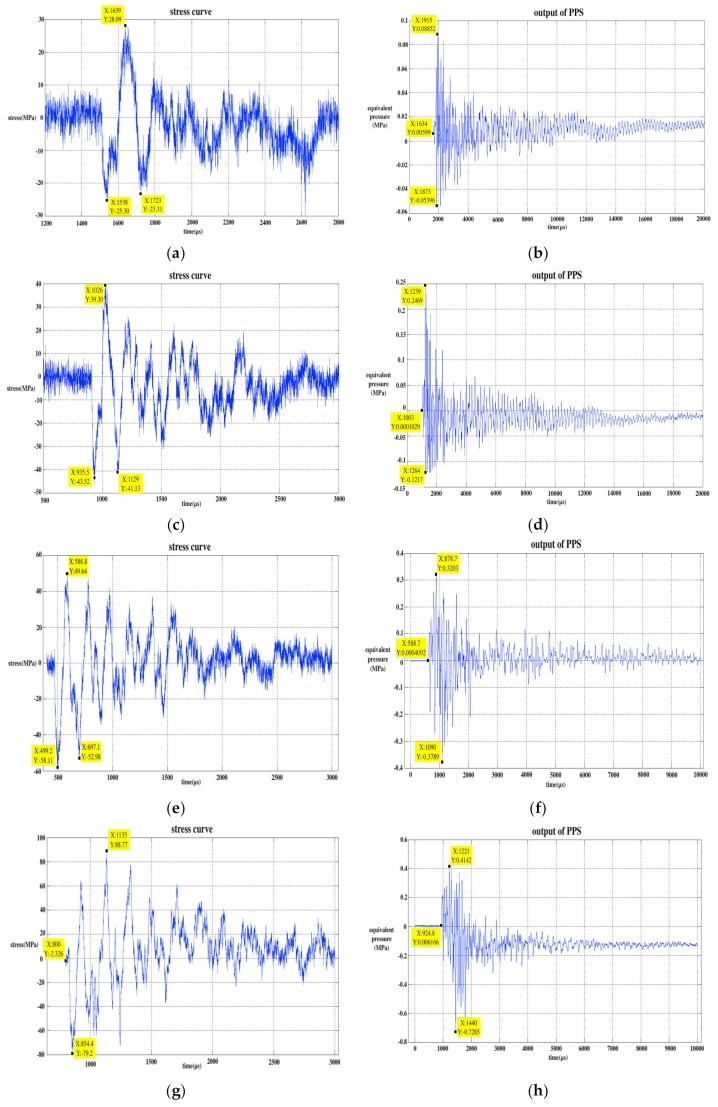
Experimental data of the stress wave effect SWE: (**a**) stress at 30 degrees; (**b**) output of PPS at 30 degrees; (**c**) stress at 60 degrees; (**d**) output of PPS at 60 degrees; (**e**) stress at 90 degrees; (**f**) output of PPS at 90 degrees; (**g**) stress at 120 degrees; (**h**) output of PPS at 120 degrees; (**i**) stress at 150 degrees; (**j**) output of PPS at 150 degrees; (**k**) stress at 180 degrees; (**l**) output of PPS at 180 degrees.

**Figure 4 sensors-20-02397-f004:**
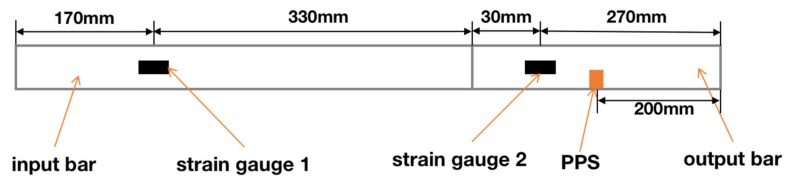
Geometric position relationship between the strain gauges and bars. The line between the two bars is the standard line of the geometric distance.

**Figure 5 sensors-20-02397-f005:**
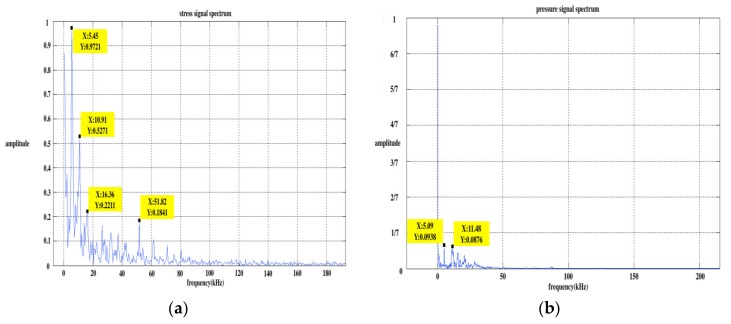
Spectrum of two signals at 120 degrees: (**a**) spectrum of the stress wave signal; (**b**) spectrum of the PPS output signal.

**Figure 6 sensors-20-02397-f006:**
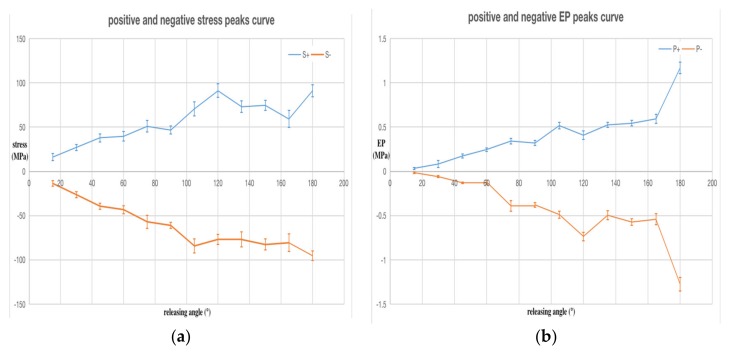
Analysis results of the SWE experimental data: (**a**) stress; (**b**) equivalent pressure (EP) of the PPS output; (**c**) EP–stress ratio. The ordinate unit of (c) is one.

**Figure 7 sensors-20-02397-f007:**
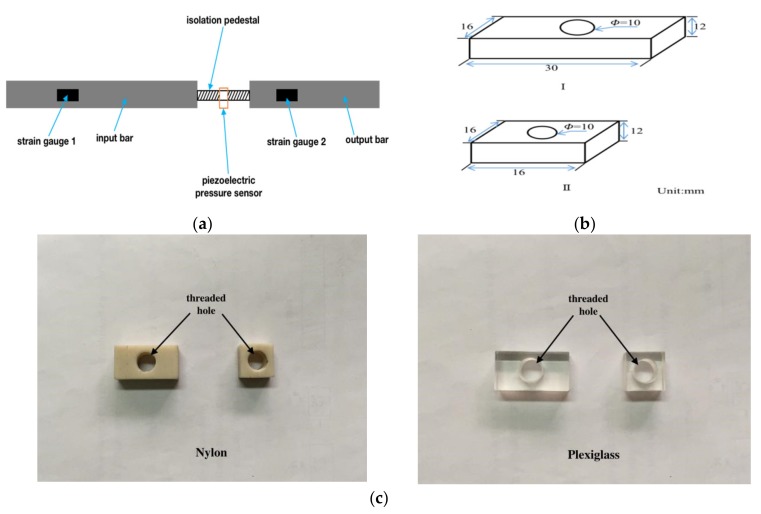
Experiment of stress wave isolation: (**a**) local structure diagram of the experiment; (**b**) the isolation pedestal dimensions (all dimensions in mm); (**c**) photograph of the isolation pedestals.

**Figure 8 sensors-20-02397-f008:**
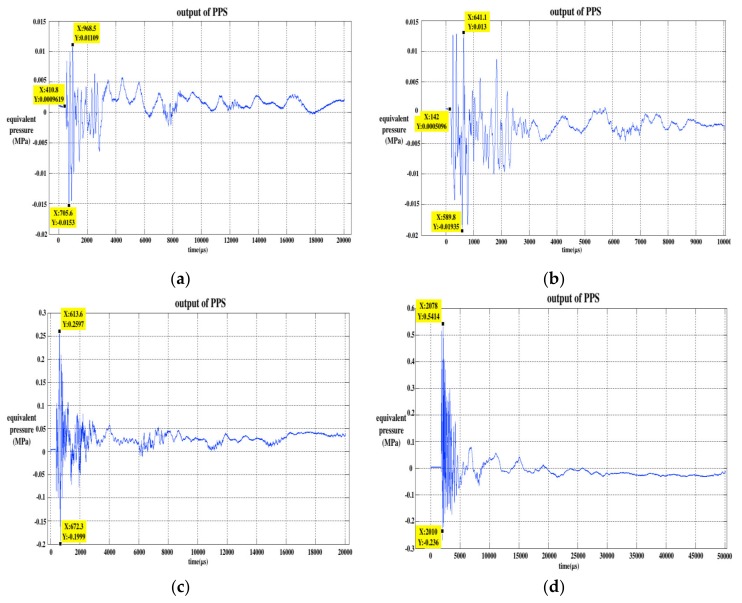
Output of the PPS for the stress wave isolation experiment at 120 degrees: (**a**) 16 mm nylon isolation pedestal; (**b**) 30 mm nylon isolation pedestal; (**c**) 16 mm Plexiglass isolation pedestal; (**d**) 30 mm Plexiglass isolation pedestal.

**Figure 9 sensors-20-02397-f009:**
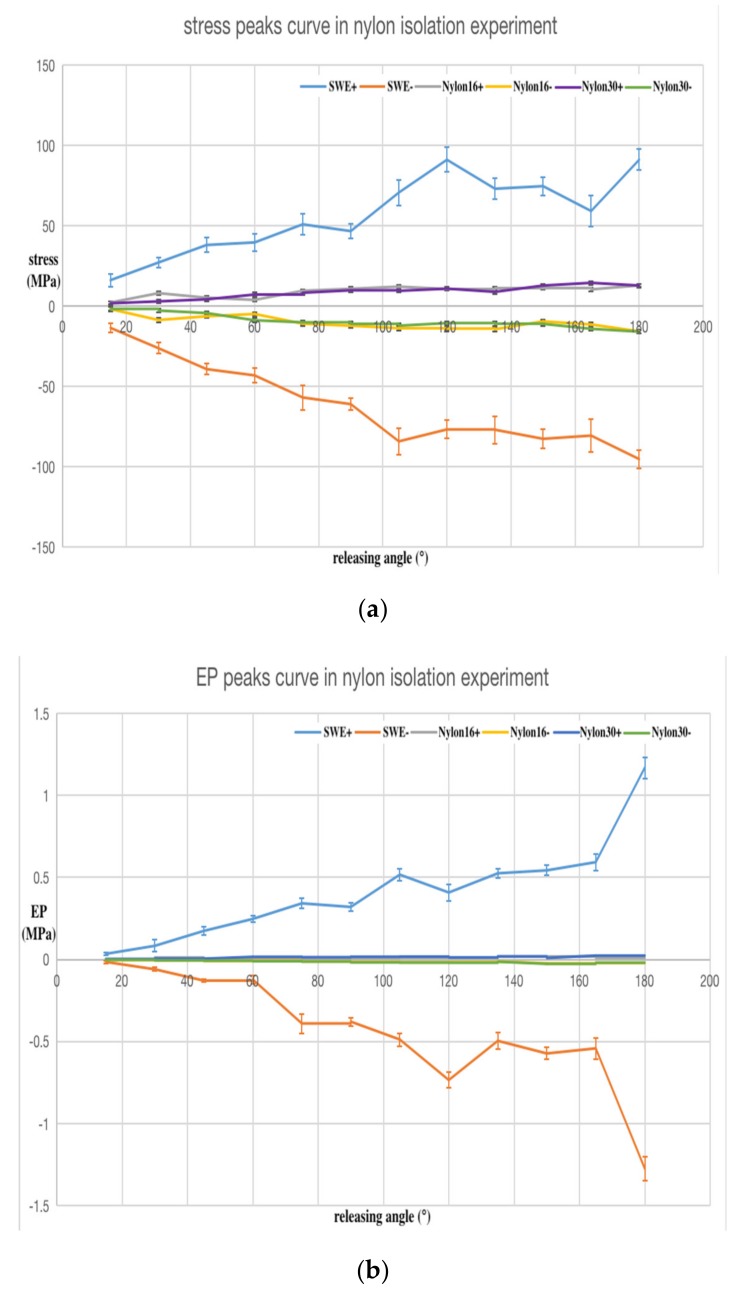
Analysis results of the nylon isolation pedestal experimental data: (**a**) stress; (**b**) EP of the PPS output; (**c**) larger view of (b); (**d**) EP–stress ratio. The ordinate unit of (d) is one.

**Figure 10 sensors-20-02397-f010:**
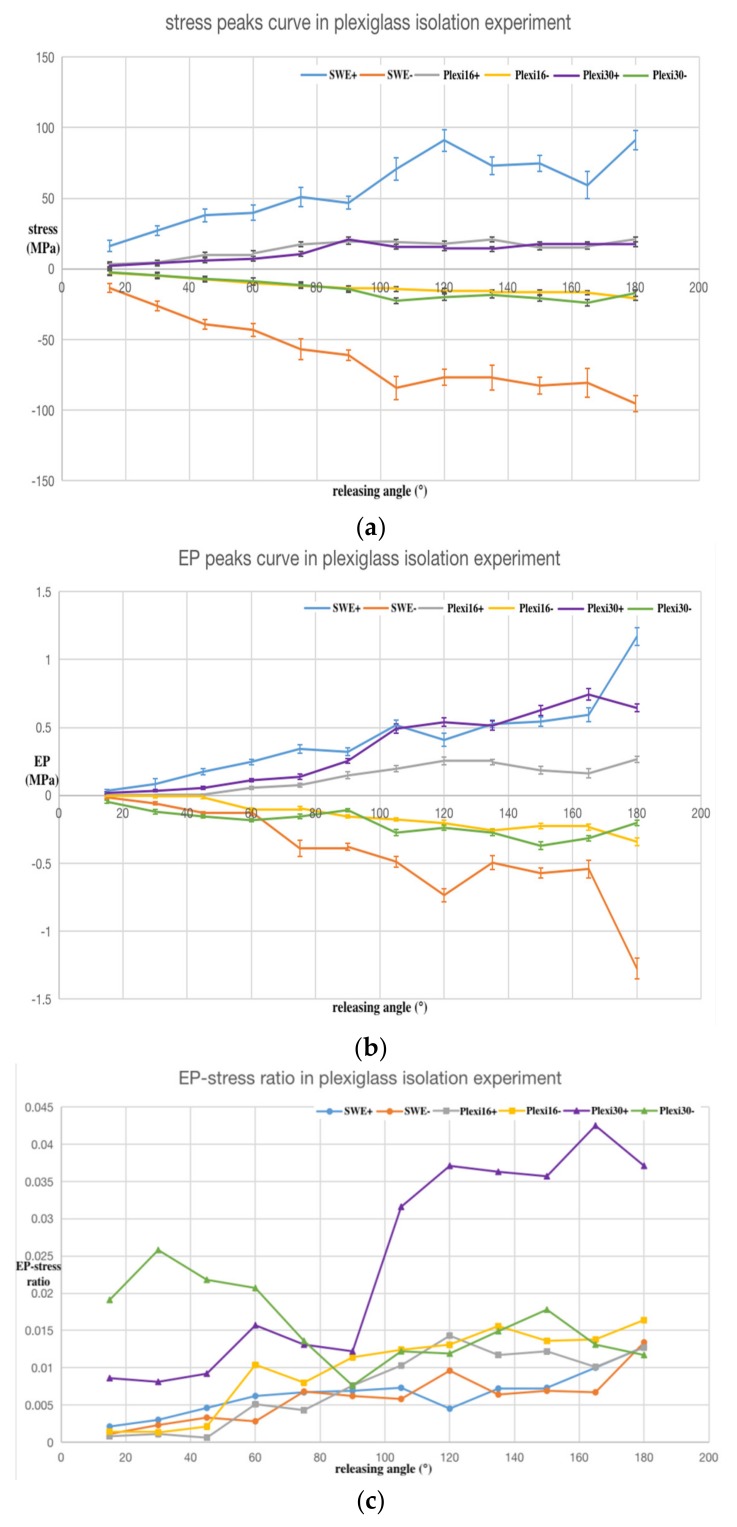
Analysis results of the Plexiglass isolation pedestal experimental data: (**a**) stress; (**b**) EP of the PPS output; (**c**) EP–stress ratio. The ordinate unit of (c) is one.

**Figure 11 sensors-20-02397-f011:**
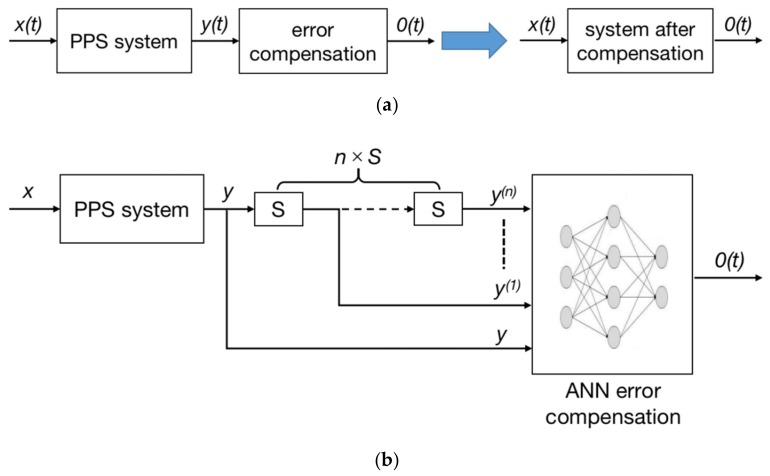
Compensation principle and algorithm block diagram: (**a**) basic compensation principle; (**b**) algorithm structure diagram.

**Figure 12 sensors-20-02397-f012:**
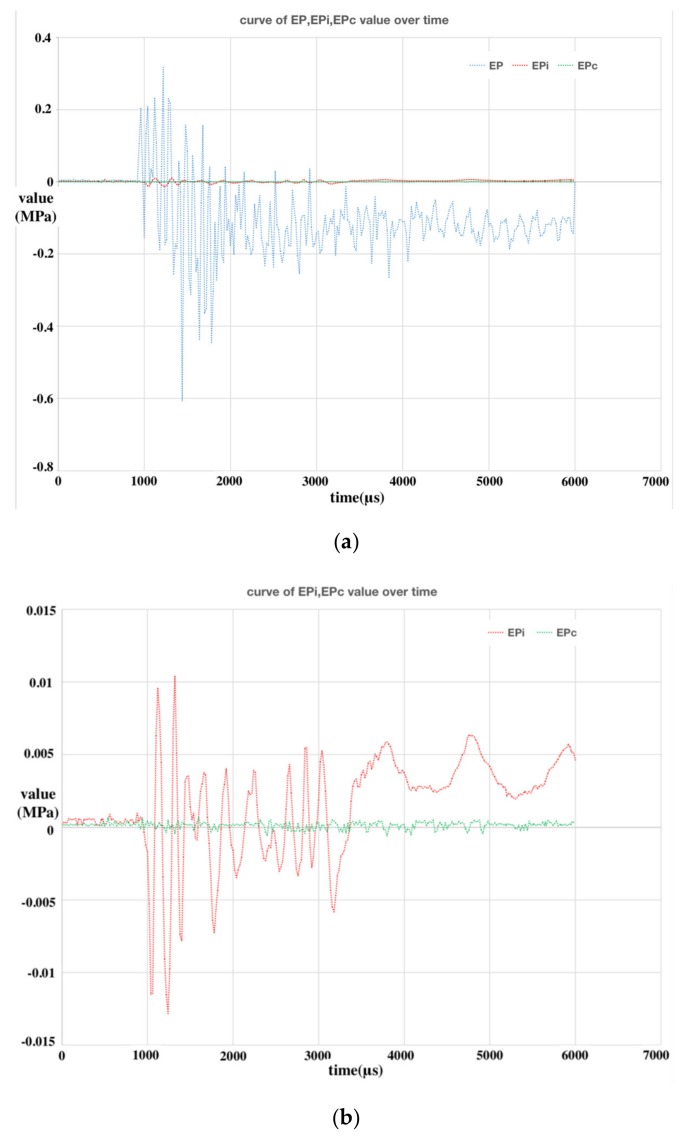
Results analysis of the back propagation neural network (BPNN) error compensation: (**a**) comparing curve of EP, EPi, and EPc at 120 degrees; (**b**) comparing curve of EPi, and EPc at 120 degrees; (**c**) comparison of the maximum peak lines at different angles; (**d**) is a larger view of (c).

**Table 1 sensors-20-02397-t001:** Stress wave effect (SWE) experimental data.

Releasing Angle (°)		15	30	45	60	75	90	105	120	135	150	165	180
**Stress Wave**	highest positive peaks (MPa)	16.10	27.09	38.00	39.60	50.88	46.64	70.67	91.10	73.04	74.65	59.17	91.11
	highest negative peaks (MPa)	13.60	26.30	39.31	43.22	56.98	61.11	84.30	76.87	77.07	82.72	80.72	95.36
	spectrum points (kHz)	6.0	6.2	5.7	5.6	5.9	5.8	5.6	5.4	5.6	5.6	5.5	5.5
		10.9	10.6	11.7	11.2	11.2	10.8	10.7	10.9	10.8	10.8	10.9	10.6
**PPS Output**	highest positive peaks (MPa)	0.034	0.083	0.174	0.247	0.342	0.320	0.517	0.408	0.525	0.543	0.593	1.169
	highest negative peaks (MPa)	0.015	0.060	0.129	0.122	0.390	0.379	0.488	0.735	0.496	0.573	0.542	1.275
	spectrum points (kHz)	5.0	5.0	5.0	5.0	5.1	5.0	5.1	5.1	5.0	5.1	5.0	5.2
		10.9	11.4	11.3	11.5	11.5	11.5	11.6	10.5	11.3	11.6	11.5	11.9
**EP–** **Stress Ratio**	positive	0.21%	0.30%	0.46%	0.62%	0.67%	0.69%	0.73%	0.45%	0.72%	0.73%	1.00%	1.28%
	negative	0.11%	0.23%	0.33%	0.28%	0.68%	0.62%	0.58%	0.96%	0.64%	0.69%	0.67%	1.34%

**Table 2 sensors-20-02397-t002:** Wave impedance parameters of each material in the experiment.

Part Name	Input Bar and Output Bar	Isolation Pedestal	
**Material**	Steel	Plexiglass	Nylon
**Wave impedance/(Pa·s·m^−1^) × 10^5^**	452	31	29

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
