# Peer review of "Experimental Study Comparing the Effectiveness of Physical Isolation and ANN Digital Compensation Methodologies at Eliminating the Stress Wave Effect Error on Piezoelectric Pressure Sensor"

_sensors, 2020, doi:10.3390/s20082397_

Round 1
Reviewer 1 Report
This paper addresses the problem of Stress Wave Effect (SWE), the shock wave overpressure originated from an explosion, on sensors. The problem is that SWE induces undesired vibrations and thus interference on pressure sensors. To tackle this issue, the authors proposed an experimental approach consisting of an isolating pedestal for sensors. Two material options for such pedestal were evaluated: nylon and plexiglass. A testbed was deployed and experimental results indicate that both options reduce the SWE. Nevertheless, the nylon approach is better because it reduces the equivalent pressure (EP)-stress and filters high frequency components. In addition, an artificial neural network (ANN) approach was implemented as a third option for compensating the SWE measuring error. Results shows that the ANN "digital" approach is even better than the pedestal "physical" approach. Yet, it does not protect the sensor.
Overall, the paper addresses an interesting topic on pressure sensor protection and error compensation methods. It is a good match for the journal’s scope. The paper is well-written (some typos, see section B), it is easy to read and to follow. I recommend a major revision addressing the following remarks:
A. Content
- State of the Art missing. The introduction explains the SWE and overviews the internal operation of a pressure sensor (then detailed in Section 2). There is no literature review of the related work and the achievements obtained so far that allow to better appreciate the contribution of this work.
- Fig 1b. It would be better to display both theoretical and actual overpressure curves in the same plot. This would allow to compare magnitudes (the 12% increase mentioned in the conclusion).
- Fig 3. It is difficult to condense (and then retain in mind) all 6x2 plots to draw a conclusion. I recommend to either add another plot showing degree VS stress / PPS or a summary table.
B. Format
25: SHPB should be spelled out in the keyword section (just as PPS and SWE)
44: the way to excludes them --> to exclude them
50: SWE experiment data --> experimental data
51: The results after compensating were --> after compensation
97: this section discussed --> discusses
166: Therefor --> Therefore
169: Figure 2(a) is the schematic diagram --> Figure 2(a) shows the schematic diagram
210, 263,341, 343,359, 392: The use of verbs in infinitive at the beginning of the sentence seems like a “receipt” that one must follows:
Release the hit bar, Repeat the test, draw the trend, etc..
Please adjust to keep a narrative style.
254: Error! Reference source not found.5.
312: Take all the factors --> taking all
417: didn’t --> did not
473: BP neural network, specify that BP stands for “back propagation”
481: A appropriate number --> An appropriate
Author Response
Response to Reviewer 1 Comments
Point 1: (A. Content) State of the Art missing. The introduction explains the SWE and overviews the internal operation of a pressure sensor (then detailed in Section 2). There is no literature review of the related work and the achievements obtained so far that allow to better appreciate the contribution of this work.
Response 1: The current researches in this field mainly focus on the errors caused by high-g impact and vibration to the PPS output signal. I have supplemented representative research in the introduction. See line 53-57 (WORD version).
Point 2: (A. Content) Fig 1b. It would be better to display both theoretical and actual overpressure curves in the same plot. This would allow to compare magnitudes (the 12% increase mentioned in the conclusion).
Response 2: Fig 1b is copied from reference and there is no data to plot the curve. In addition, the theoretical curve contains no coordinate values, but only to characterize the curve qualitatively. Therefore, it is a pity that the two curves cannot be displayed in the same plot. The 12% increase is irrelevant here, it refers that maximum EP have reached 12% of the PPS range.
Point 3: (A. Content) Fig 3. It is difficult to condense (and then retain in mind) all 6x2 plots to draw a conclusion. I recommend to either add another plot showing degree VS stress / PPS or a summary table.
Response 3: I have supplement SWE experimental data in table 1 in the manuscript. See line 274-275 (WORD version). And combining the analysis results of SWE experiment shown in figure 6, the conclusion can be drawn.
Point 4: (B. Format) 25: SHPB should be spelled out in the keyword section (just as PPS and SWE)
Response 4: I have spelled out SHPB as the reviewer requested. See line 26-27 (WORD version).
Point 5: (B. Format) 44: the way to excludes them --> to exclude them
Response 5: I have revised as the reviewer requested. See line 51 (WORD version).
Point 6: (B. Format) 50: SWE experiment data --> experimental data
Response 6: I have revised as the reviewer requested. See line 63 (WORD version).
Point 7: (B. Format) 51: The results after compensating were --> after compensation
Response 7: I have revised as the reviewer requested. See line 64 (WORD version).
Point 8: (B. Format) 97: this section discussed --> discusses
Response 8: I have revised as the reviewer requested. See line 113 (WORD version).
Point 9: (B. Format) 166: Therefor --> Therefore
Response 9: I have revised as the reviewer requested. See line 184 (WORD version).
Point 10: (B. Format) 169: Figure 2(a) is the schematic diagram --> Figure 2(a) shows the schematic diagram
Response 10: I have revised as the reviewer requested. See line 188 (WORD version).
Point 11: (B. Format) 210, 263,341, 343,359, 392: The use of verbs in infinitive at the beginning of the sentence seems like a “receipt” that one must follows:
Release the hit bar, Repeat the test, draw the trend, etc..
Please adjust to keep a narrative style.
Response 11: I have revised as the reviewer requested. See line 245, 302, 381, 383, 384, 399, 434 (WORD version).
Point 12: (B. Format) 254: Error! Reference source not found.5.
Response 12: I have checked that this error occurred when WORD is converted into PDF, and the WORD version does not have this error. See line 293 (WORD version).
Point 13: (B. Format) 312: Take all the factors --> taking all
Response 13: I have revised as the reviewer requested. See line 351 (WORD version).
Point 14: (B. Format) 417: didn’t --> did not
Response 14: I have revised as the reviewer requested. See line 460 (WORD version).
Point 15: (B. Format) 473: BP neural network, specify that BP stands for “back propagation”
Response 15: I have revised as the reviewer requested. See line 514 (WORD version).
Point 16: (B. Format) 481: A appropriate number --> An appropriate
Response 16: I have revised as the reviewer requested. See line 523 (WORD version).
Reviewer 2 Report
See attached file

Author Response
Response to Reviewer 2 Comments
Point 1: (line 13-14 in PDF version, the same below) Add majuscules
Response 1: I have revised as the reviewer requested. See line 13-14 (WORD version).
Point 2: (line 15-16) Add majuscules
Response 2: I have revised as the reviewer requested. See line 15-16 (WORD version).
Point 3: (line 21) Not defined
Response 3: ANN means artificial neural network here. I have supplemented in the manuscript. See line 21 (WORD version).
Point 4: (line 32) Even without fragment hit, the shockwave arriving on metal probe produce shockwave against the metal. As the sound velocity is higher than in air, stress arrive on the sensor by the probe before arriving by air.
Response 4:
It is right that stress wave will be produced in the metal probe when the shock wave in air hit it. However the stress value produced by air hit is smaller than that by solid fragment hit. And the PPS is intended to measure the shock wave pressure in air, PPS cannot measure the stress in the probe, so the stress in the probe is an interference.
In addition, the pen-shape is designed to reduce the effect of air shock wave on the solid pressure sensor, and can maintain the stability of the shock wave flow field during the overpressure measurement. In a word, the stress produced by the fluid on the solid is not obvious when using the pen-shaped pressure sensor.
But when the sensors are in other shapes, the situation is less clear. The shape of the sensor may cause a complex reaction to the air shock wave.
However, as an objective factor, it is necessary to explain it in the introduction. So I have supplemented the description about this phenomenon in the manuscript. See line 33 (WORD version).
Point 5: (line 59-60) It is not entirely true for side-on configuration (when shockwave front is perpendicular to sensor surface) : with a shockwave speed in air of 1000m/s and a sensing surface of 1 mm of diameter, the delay is 1 µs.
Response 5: It is right that the statement in the manuscript is not quite accurate when the shock wave is perpendicular to the sensor. My intention is to emphasize that the pressure sensor can be thought of as a whole when it is equivalent to a second-order mechanical system. But there is ambiguity here. So I have revised in the manuscript. See line 72-73 (WORD version).
Point 6: (line 71) Nozzel probes are generally used in side-on configuration (probe length perpendicular to shockwave front) reducing the probability of hit. But it still have the problem of wave propagation in the metal after shockwave (see remark line 32)
Response 6: I have supplemented the description about this phenomenon in the manuscript. See line 82 (WORD version).
Point 7: (line 77-78) see remark line 32
Response 7: Theoretically, it is right that stress wave will be produced in the metal probe when the shock wave in air hit it. However, the density of air is obviously less than that of solid metal, so the magnitude of stress generated by its impact is difficult to reach the degree shown in the Figure 1(b). So, the expression "most likely" is used in the manuscript to speculate that the interference with such a large value may be caused by the impact of the solid fragments. See line 89 (WORD version).
Point 8: (line 83-84) Bad quality of figures.
Text size to small.
Give references for figures copied from other authors.
Fig 1a : Usually for side on configuration, the nozzle probe is perpendicular to shockwave front.
Fig 1b : Precise that it is for side on configuration.
Fig 1c : Precise where is the sensing part of PPS. "Diaphragm" : Drilled to allow air passing.
Response 8: I have added reference for figures (see line 101) and revised the description of the Fig 1a and Fig 1b (see line 98-99). Fig 1c shows the internal structure of PPS, and the “Diaphragm” is the surface to sense the air pressure. Then the pressure is transferred into the piezoelectric element (“quartz plates” in Fig 1c). There is no need to drill in the sensor.
Point 9: (line 97) Precise : after hitting by a particule
Response 9: I have added the precise description in the manuscript. See line 113-114 (WORD version).
Point 10: (line 170) Make a new sentence
Response 10: I have revised as the reviewer requested. See line 189 (WORD version).
Point 11: (line 171) Explain the contact between the input and the output bar.
Response 11: I have supplemented the description about the contact between two bars. See line 190-193 (WORD version).
Point 12: (line 172) I don't understand what is the strain rate
Response 12: Strain rate is the change in strain with respect to time, being t the time. Many materials’ constitutive relation equation is related to strain rate especially the viscoelastic material. We know that the conventional use of SHPB is testing the mechanical properties of the materials. So strain rate is an important technical indicator of SHPB. See the newly added reference [19] line 627-628 in WORD version.
Point 13: (line 172) Add spaces
Response 13: I have revised as the reviewer requested. See line 193-194 (WORD version).
Point 14: (line 173) length l
Response 14: I have revised as the reviewer requested. See line 195 (WORD version).
Point 15: (line 174) diameter d
Response 15: I have revised as the reviewer requested. See line 196 (WORD version).
Point 16: (line 175) Make new sentence
Response 16: I have revised as the reviewer requested. See line 199 (WORD version).
Point 17: (line 176) Precise which strain gauges is used and how it is fixed to the bar.
Response 17: I have make a new paragraph to explain the usage of strain gauges compared this paper to conventional material mechanical properties test. See line 200-208 (WORD version).
Point 18: (line 179) Make new sentence. Describe where the PPS is placed.
Response 18: I have made a new sentence as the reviewer requested. See line 211 (WORD version). Supplemented the description about the mounting position of the PPS in line 214 (WORD version).
Point 19: (line 180) Make new sentence
Response 19: I have revised as the reviewer requested. See line 212 (WORD version).
Point 20: (line 181-182) Precise that the sensitive surface of PPS is not in contact with the bar (Ref Fig 1c : cavity between "membrane" and "sensitive surface"). Not clear in fig 2a)
Response 20: I have supplemented the description about this question in the manuscript and supplemented some details about the contact between PPS and the space with atmosphere in fig 2a. See line 215-216, 218-219 (WORD version). Ref response 8.
Point 21: (line 194) Start a new paragraph
Response 21: I have revised as the reviewer requested. See line 229 (WORD version).
Point 22: (line 199) Modify the previous sentence to include this part.
Response 22: I have merged the sentence “ plastic deformation occurs in materials ” into the previous paragraph. See line 230-232 (WORD version).
Point 23: (line 202) Add space : Y = 300 MPa
Response 23: I have revised as the reviewer requested. See line 236 (WORD version).
Point 24: (line 202) Add spaces. What are these 2 values. You can reduce the precision of the numbers.
Response 24: is the wave impedance of one material. It is the product of the density and the sound velocity of the material. Wave impedance is useful in Equation 9 to calculate the reflection and transmission coefficient between two different materials. And wave impedance is also used to calculate the critical velocity when plastic deformation occurs in material through Equation 10. I have added spaces and reduced the precision of the numbers as the reviewer requested. See line 236 (WORD version).
Point 25: (line 203) Add spaces. You can reduce the precision of the numbers.
Response 25: I have added spaces and reduced the precision of the numbers as the reviewer requested. See line 237 (WORD version).
Point 26: (line 204) 15.9 m/s
Response 26: I have revised as the reviewer requested. See line 239 (WORD version).
Point 27: (line 210-211) Problem on the sentence
Response 27: I have rewritten the sentence in the manuscript. See line 245-246 (WORD version).
Point 28: (line 213) Add spaces.
Response 28: I have revised as the reviewer requested. See line 248 (WORD version).
Point 29: (line 229) Text illegible in the figures. Take the same time reference for stress and PPS output. Stress for strain gauge 1 or strain gauge 2 ?
Response 29:
In order to reduce the length of the manuscript, the figures are set as small size. But it’s easy to see the text clearly when figures are zoom in.
PPS output signals lag behind the stress wave signal. The time references of the two types of signals are different, and this paper focus on the amplitude relationship between the two types of output signals. And it is a good idea for further research on the dynamic mechanism of stress wave acting on PPS. However, it is necessary to set up the synchronous trigger mechanism on the test equipment to collect the two signals with same time reference. This can be used as a technical scheme for further research.
Stress wave signal of SWE experiment is collected by strain gauge 2 on the output bar (ref response 17, line 200-208 in the manuscript). I have supplemented the description in the manuscript. See line 249-250 (WORD version).
Point 30: (line 236) what do you mean ?
Response 30: My intention is to emphasize that the stress wave curve oscillates more violently with the increase of the releasing angle. This indicates an increase in the high-frequency component of the stress wave signal. So I have revised the statement in the manuscript. See line 271 (WORD version).
Point 31: (line 239) It is the velocity of sound in steel ? Put the value (5800 m/s) here.
Then it is an acoustic wave propagation and not a shock wave propagation (velocity higher than sound). It is only because the the impact velocity is not very high ?
You have to explain this point in the right place in the paper.
Response 31: Yes, it is the velocity of sound in steel. See line 278 (WORD version). This paper is a research on the pressure sensor used in overpressure measurement. PPS is intended to measure the overpressure of shock wave in the air. Because shock wave and fragments have limited impact energy on the PPS, the stress wave generated by the impact can be considered as sound wave in the solid rather than shock wave. This is because generating a shock wave requires the disturbance in the medium behind to keep up with the disturbance ahead. Obviously the impact in this paper will not produce a shock wave in solid. It takes a lot of burst energy to create a shock wave in solid. It has been reported that laser micro-explosion method can generate shock wave in metal. I have supplemented the description in the manuscript. See line 39-44 (WORD version).
Point 32: (line 241-245) Not clear. Explain that you have 2 bar separated by ?? With reflections on each interface.
Response 32:
The experiment equipment in this paper is made of two separated bars. When doing the SWE experiment, the two bars is in contact with each other, and the PPS is installed on the output bar. When doing the stress wave isolation experiment in the below, the PPS is installed on the isolation pedestal which is sandwiched by the input bar and output bar. These are two different experiments and different use of the experiment equipment.
Relating to the SWE experiment, there is no reflection on the interface, because the material and cross-section of the two bars are same (ref response 11).
Point 33: (line 247) Precise what is the line 30mm before strain gauge 2 : Contact surface between input and output bar. Add PPS sensor.
Response 33: The line is the standard line of the geometric distance in the figure 4, corresponding to the contact surface of the two bars in actual. See line 285-286 (WORD version). I have added PPS in the fig 4. See line 284 (WORD version).
Point 34: (line 249)
Response 34: I have deleted the sentence in the manuscript (see line 287-288) and put the value (5800 m/s) in line 278. See Response 31.
Round 2
Reviewer 1 Report
The paper has undoubtedly improved from its previous version. I appreciate that my remarks were taken into account. In particular, Table I eases the understanding of the SWE experimental data in Fig. 3. I have also read the comments raised by my colleague reviewer and the answers provided. I consider that our main concerns have been correctly addressed. I have no further content comments or mayor remarks; therefore I recommend now this paper’s acceptance.
Minor:
a) Line 21: Finally, the method based on Artificial Neural Networks (ANN) which trained by using the data of SWE study was applied to compensate the SWE error of the PPS output signal.
Rephrase to: Finally, an Artificial Neural Network (ANN) was trained with the data of the SWE study and was further applied to compensate the SWE error of the PPS output signal.
b) Line 42: Obviously --> Evidently
Author Response
Point 1:
Line 21: Finally, the method based on Artificial Neural Networks (ANN) which trained by using the data of SWE study was applied to compensate the SWE error of the PPS output signal.
Rephrase to: Finally, an Artificial Neural Network (ANN) was trained with the data of the SWE study and was further applied to compensate the SWE error of the PPS output signal.
Response 1: I have rewritten the sentence as the reviewer recommended. See line 21-23 (WORD version).
Point 2:
Line 42: Obviously --> Evidently
Response 2: I have revised as the reviewer requested. See line 44 (WORD version).
Reviewer 2 Report
see file

Author Response
Thank you very much for your review comments. Please find our response at the attachment.
